# Multi-Omics Analysis Reveals Intricate Gene Networks Involved in Female Development in Melon

**DOI:** 10.3390/ijms242316905

**Published:** 2023-11-29

**Authors:** Zhongyuan Wang, Vivek Yadav, Xiaoyao Chen, Siyu Zhang, Xinhao Yuan, Hao Li, Jianxiang Ma, Yong Zhang, Jianqiang Yang, Xian Zhang, Chunhua Wei

**Affiliations:** State Key Laboratory of Crop Stress Biology in Arid Areas, College of Horticulture, Northwest A&F University, Xianyang 712100, China; zydx@nwafu.edu.cn (Z.W.); vivekyadav@nwafu.edu.cn (V.Y.); xiaoyaochen7186@163.com (X.C.); zhangsiyu@nwafu.edu.cn (S.Z.); yuanxinhao@nwafu.edu.cn (X.Y.); yuanyilihao123@163.com (H.L.); majianxiang@126.com (J.M.); zhangyong123@nwsuaf.edu.cn (Y.Z.); yangjq1208@126.com (J.Y.)

**Keywords:** cucurbits, ethylene, omics, sex differentiation, female development

## Abstract

Sexual differentiation is an important developmental phenomenon in cucurbits that directly affects fruit yield. The natural existence of multiple flower types in melon offers an inclusive structure for studying the molecular basis of sexual differentiation. The current study aimed to identify and characterize the molecular network involved in sex determination and female development in melon. Male and female pools separated by the F_2_ segregated generation were used for sequencing. The comparative multi-omics data revealed 551 DAPs and 594 DEGs involved in multiple pathways of melon growth and development, and based on functional annotation and enrichment analysis, we summarized four biological process modules, including ethylene biosynthesis, flower organ development, plant hormone signaling, and ubiquitinated protein metabolism, that are related to female development. Furthermore, the detailed analysis of the female developmental regulatory pathway model of ethylene biosynthesis, signal transduction, and target gene regulation identified some important candidates that might have a crucial role in female development. Two CMTs ((cytosine-5)-methyltransferase), one AdoHS (adenosylhomocysteinase), four ACSs (1-aminocyclopropane-1-carboxylic acid synthase), three ACOs (ACC oxidase), two ARFs (auxin response factor), four ARPs (auxin-responsive protein), and six ERFs (Ethylene responsive factor) were identified based on various female developmental regulatory models. Our data offer new and valuable insights into female development and hold the potential to offer a deeper comprehension of sex differentiation mechanisms in melon.

## 1. Introduction

Fruit development and seed formation in plants are directly linked with flowering patterns. Many plant species naturally exist with perfect flowers. Our current understanding of natural sex differentiation in many plants is not extensive. The mechanism of sex differentiation and development is limited because many model plants are not dioecious in nature [1,2]. Sexual differentiation in plants is crucial for reproduction. In practice, the economic values often differ between the existence of male and female flowers. Female flowers can produce fruits and seeds, while male flowers are the only source of pollen to fertilize female flowers [3]. Understanding sexual differentiation is essential in plant breeding programs, enabling the development of cultivars with desirable traits [3,4]. The first transcriptome-based study for sex-related genes was carried out in *Silene latifolia*, where dosage compensation in the sex chromosome was identified [5]. Transcriptome-based studies were carried out in many plants, including some cucurbits, to reveal the genetic mechanisms involved in sex differentiation and flower development. For instance, transcriptome profiling of the flower buds of *Coccinia grandis* (L.) Voigt was performed by Mohanty [6]. The results revealed that many hormone-responsive genes, including ADP-ribosylation factor 6 (ARF6), 1-Aminocyclopropane-1-carboxylic acid synthase (ACC/ACS), sucrose non-fermenting-1-related protein kinase 2 (SNRK2), and BRI1-associated receptor kinase 1 (BAK1), were involved in flower differentiation [6].

The Cucurbitaceae family is composed of 120 genera and 960 species, which include many economically important cultivated species [4]. The Cucurbitaceae family holds significant agricultural and nutritional importance due to its diverse members, including cucumbers, pumpkins, and watermelons [1,7,8,9]. These plants are valued for their economic contributions as staple food crops and sources of essential nutrients. In melon, some studies have shown important information about floral sex differentiation and development. For example, the mechanisms regulating the bisexual flowers in melon by transcriptome showed that the ethylene downstream gene *CmERF1* significantly contributes to bisexual bud formation [10]. In another study, bulk segregant populations were utilized for transcriptome analysis, which revealed ethylene, abscisic acid (ABA), auxin/indole-3-acetic acid (IAA), and aminocyclopropanecarboxylate (ACC) oxidase were involved in different sex type expressions, suggesting that they might have important roles in melon sex determination [11]. The transcriptome studies provided some reliable findings regarding in-depth information about sex forms in many crops. However, there are some limitations in transcriptome-based studies, such as that gene expression abundance may not be reliable and not always precisely reflect the proteins coding the phenotypic character or biological activities [12,13]. Therefore, post-transcriptional modification should be validated through proteomics studies [14,15].

The most common sex forms in cultivated melon plants are andromonoecious, where bisexual flowers are only found on the leaf axil of lateral branches and male flowers on the main stem and lateral branches [10,16]. The fruit development is from bisexual flowers in plants [17]. Melon flowers serve as a model system for studying physiological and molecular aspects of sex determination in plants due to the availability of different types of stable sex forms in melon species. Moreover, the flowering pattern in melons also provides an ideal system of choice for researchers to study the biological process from flowering to fruit development [18]. Based on the distribution of flowers of different sexes on the same plant, melons can be classified as andromonoecy, androecy, gynoecy, hermaphrodite, and monoecy [19]. In melon, all four whorls of floral organ primordia emerge at the preliminary stage of flower development. However, due to the function of the sex-determining gene, one of the sex organ primordia (stamen or carpels) stagnates at a certain stage, resulting in abortion and loss of function in this sex organ and leading to bisexual, female, and male flowers in melon [20]. The floral primordia of melon flowers are initially bisexual, and the differentiation in sex forms occurs at lateral stages of development [21]. The restricted development of the carpel or the stamen results in a female flower or a unisexual male flower. The research reports to date suggest that sex differentiation in melon is mainly regulated by a set of genes that includes *CmACS7*, *CmWIP1*, and *CmACS11* [21,22,23]. It was evidenced that the expression of *CmACS7* is limited to the carpel of bisexual and female flowers, which makes it an important gene that regulates sex differentiation in melon [22]. Further studies revealed restricted expression of *CmACS7* in the male flower, and it inhibits stamen development in the female flower of melon. Some reports also revealed that mutations in *CmACS7* can result in the formation of bisexual or male flowers [22]. On the other hand, *CmWIP1* acts as an inhibitor that suppresses female organ development through a complex network where it has a strong interaction with *CmbZIP48* [24]. In *CmWIP1* mutant plants, only bisexual or female flowers developed. Some research results revealed that *CmWIP1* can suppress the expression of *CmACS7* because it works as an upstream suppressor on male flowers [20,24]. Similarly, *CmACS11* has an important role at an upstream level that restricts the expression of *CmWIP1*. The expression of the *CmACS11* gene was not detected in male flowers; rather, the expression was detected in vascular bundles of female flowers or bisexual flowers. Therefore, it was assumed that *CmACS11* plays a critical role in female sex organ development in melon flowers [21]. The mutant of *CmACS11* showed a generation of androecy sex forms in flowers. In addition to the genetic network controlling sex expression in cucurbits, phytohormones, including gibberellin (GA) and ethylene (ET), profoundly affect flower expression in cucurbits [25]. External factors such as temperature, photoperiod, and ethylene-controlling agents also regulate flowering patterns in cucurbits [26]. The ET-based pathways are thoroughly studied in many cucurbits because they significantly regulate sex expression [8,27]. For instance, the development of female flowers required more ethylene than what is required for male flowers in watermelon [1]. Ethephon and IAA applications in pumpkins significantly increased the number of female flowers [28]. Similarly, the application of an ethylene inhibitor resulted in a higher number of male flowers in cucumber [29]. In the case of melon, the external application of ethephon treatment boosts the number of bisexual and female flowers [16].

However, it is quite possible that other genes, complex genetic networks, and nongenetic factors could contribute to sex expression [11]. The application of multi-omics can lead to some concrete conclusions and contribute to more detailed information on sex expression research in cucurbit crops. The main mechanisms and some candidate pathways that regulate sex differentiation in melon are controlled by several mechanisms, and some important mechanisms are still elusive and require further research [30]. Therefore, the current study was designed to decipher the gene networks that may contribute to female flower development. We mapped the comparative transcriptome and total proteome profiling of male and female dominant flower bud pools generated from crossing in melon plants. In the current study, stable ‘0426’ (monoecious) and ‘Y101’ (hermaphrodite line) were used as parent material because of their distinct flowering characteristics [26]. With the advancement of genomic techniques, we have more opportunities to link the available information with previously identified results. The current research provides important data about molecular mechanisms involved in female flower development in plants. Transcriptome data supported by proteome analysis of male and female dominant pools evidenced an intricate gene network that contributes to female flower development in melon.

## 2. Results

### 2.1. Proteome and Transcriptome Raw Data Analysis and Protein/Gene Identification

Melon sex determination primarily occurs during the early stage of flower bud development and cannot be distinguished by the naked eye. The female phenotype ‘Y101’ and the male phenotype ‘0426’ differ in the mechanism of sex differentiation. In the F_2_ population, there are two female phenotypes (gynoicous and hermaphrodite) and two male phenotypes (monoecious and andromonoecious). To eliminate the influence of individual sex type differences, two types of mixed pools (F pool and M pool) were selected for comparative analysis from the F_2_ isolated population. Due to the specific temporal and spatial nature of the sex-determining gene, the apex tissue, which is rich in flower bud primordia, was selected for proteome and transcriptome analysis.

The original data analysis of the proteome showed that a total of 41,998 peptides were obtained from the female pool (F) and the male pool (M). Among these, 7130 proteins were identified. After comparison with the melon reference gene proteome (DHL92, V4.0), a total of 39,812 (94.79%) peptide matches were found. Finally, 6870 (96.35%) melon-specific proteins were identified (Figure 1). Among the specific proteins, more than 76.27% are composed of more than two peptides (Figure 1C). The identified proteins (6870) have a wide range of molecular weight distribution, with the majority (78.05%) concentrated in the range of 10–70 kDa. Among these, proteins of size 30–40 kDa are the most abundant, with 1131 instances.

The transcriptome raw data analysis revealed that 45,179,104 and 44,568,262 clean reads were obtained from the F pool and M pool samples, respectively. The matching rate after comparison with the reference genome (DHL92, V3.6.1) was above 95%. Approximately 94% of the total matching reads were uniquely mapped to a single position in the genome (Unique map), while around 1% of the reads matched multiple positions (Multimap) in the genome. Both positive and negative matches accounted for approximately 47% each (Table 1). From the matched reads, a total of 18,222 melon genes were identified. Among these genes, 17,460 were shared between the female phenotype (F) and the male phenotype (M), while 367 genes were specific to the female phenotype and 395 genes were specific to the male phenotype (Figure 1B).

### 2.2. Identification and Analysis of Differential Proteins and Differential Genes

Among the identified proteins, the expression value of the F pool protein was higher than the expression value of the M pool protein, with a fold change value. The significance of the difference between F and M was determined using a *t*-test, resulting in a *p*-value. To screen for differentially abundant proteins (DAPs), the following criteria were applied: fold change ≥ 1.5 or fold change ≤ 0.67, and *p*-value ≤ 0.05. As a result, a total of 551 DAPs were screened from the identified proteins, out of which 434 DAPs were up-regulated and 117 DAPs were down-regulated (Figure 2A,B and Appendix A).

In this study, correlation analysis was conducted on the top 50 differentially significant proteins (ranked by *p*-value). It was found that 13 DAPs exhibited a highly negative correlation with the remaining 47 DAPs, showing opposite expression patterns. This indicates the presence of opposite mechanisms of action between these genes in melon female regulation (Figure 2C). Regarding the 18,222 genes identified by RNA-seq, the expression value of the F pool gene was compared to that of the M pool gene using fold change values, and the significance of the difference was determined using a *t*-test and a *p*-value. The screening condition used was |log2 (Fold Change)| > 0.4 and *p*-value < 0.05, resulting in a total of 594 differentially expressed genes (DEGs) identified from the gene set. Among these, 323 DEGs were up-regulated and 271 DEGs were down-regulated (Figure 2A and Appendix A).

### 2.3. Omics Data Validation Test

Out of the total identified differently expressed proteins, 17 up-regulated proteins and 3 down-regulated proteins were selected from a pool of 551 differential proteins based on descriptions in previous studies, and their roles in various pathways were studied and compared to identify candidates. Among these proteins, 7 are hormone-related proteins, including genes related to auxin and gibberellin. Additionally, there are 5 ethylene synthesis-related proteins, 2 transcription factors associated with growth and development, 2 proteins associated with flower organ development, 6 signal transduction-related proteins, and 2 membrane proteins. Figure 3 illustrates that the majority of the protein-coding genes, specifically 19 in total, display concordant patterns of transcriptional activity that correspond with their protein expression profiles. Furthermore, the female phenotypic parent ‘Y101’ and the male phenotypic parent ‘0426’ show a strong correlation with the F pool and M pool, respectively. However, it is worth noting that the gene expression level of one ABA response protein (MELO3C010850.2.1) contradicts the observed protein expression trend (Figure 3N). This finding suggests that the transcription level of a few protein-coding genes may not be fully representative of their protein abundance (Figure 3).

Similarly, 15 up-regulated genes and 5 down-regulated genes were selected from a pool of 594 differentially expressed genes based on GO, KEGG, and functional annotation analysis of identified DEGs. These include 5 genes associated with flower organ development, 2 members of the ACS family of ethylene synthetic rate-limiting enzymes, 2 genes involved in hormone response, 2 zinc finger protein-coding genes, and one transmembrane protein-coding gene. Figure 4 shows that the expression trends of all the genes in the female (F) pool and male (M) pool are consistent with the transcriptome data. Moreover, the female phenotypic parent ‘Y101′ and the male phenotypic parent ‘0426′ exhibit the same expression trend as the F pool and M pool (Figure 4).

In summary, the proteome and transcriptome data obtained in this study exhibit high confidence and accurately reflect protein/gene expression levels, thereby providing a reliable foundation for further analytical studies. Consequently, researchers can now proceed with confidence and utilize the data from both sources.

### 2.4. Differential Protein and Differential Gene Function Enrichment Analysis

GO and KEGG enrichment analyses were conducted to verify the authenticity of the data and explore the potential regulatory network and related pathways of female traits in melon. A total of 551 differential proteins and 594 differential genes were subjected to analysis. The 551 differential proteins were categorized into three groups based on their GO functions: 380 DAPs were annotated for molecular function, 297 DAPs for biological processes, and 279 DAPs for cellular components (Figure 5A). The top 30 entries in the GO annotation revealed that the largest proportion of DAPs in terms of molecular functions belonged to “ATP binding” (GO:0005524), with 92 DAPs (90 up-regulated and 2 down-regulated). This was followed by “Metal binding” (GO:0046872) and “DNA binding” (GO:0003677), with 45 (41 up-regulated, 4 down-regulated) and 31 (27 up-regulated, 4 down-regulated) DAPs, respectively. Additionally, “Zinc ion binding” (GO:0008270) showed a high proportion of 15 DAPs, all of which were up-regulated. In terms of cellular components, the differential proteins were widely distributed in various organelles, such as the nucleus, cytoplasm, mitochondria, and plasma membrane. The nucleus (GO:0005634) accounted for the largest proportion, with 59 DAPs (46 up-regulated, 13 down-regulated). Regarding biological processes, the 297 DAPs were primarily involved in protein transport (GO:0006886 and GO:0006606), energy metabolism (GO:0005975), transcription and transcriptional regulation (GO:0006351 and GO:0006355), and ubiquitinated protein metabolism (GO:0006511) (Figure 5A).

The KEGG enrichment analysis identified a total of 132 DAPs, distributed among the top 20 enrichment categories. Notably, the category of “Nucleocytoplasmic transport” (cmo03013), which encompasses the transport of molecules between the nucleus and cytoplasm, exhibited the highest representation with 24 DAPs (23 up-regulated and 1 down-regulated). Following this, the category of “Protein processing in endoplasmic reticulum” (cmo:04141), involved in the synthesis and folding of proteins, contained 20 DAPs (10 up-regulated and 10 down-regulated). Another significant category was “Ubiquitin-mediated proteolysis” (CMO:04120), which is associated with the degradation of proteins by the ubiquitin system. This category is composed of 10 DAPs, all of which were up-regulated. Furthermore, two additional categories, “Alanine, aspartate, and glutamate metabolism” (cmo:00250) and “Plant hormone signal transduction” (cmo:04075), related to the ethylene synthesis pathway, exhibited notable proportions with 7 DAPs each. Within the “Alanine, aspartate, and glutamate metabolism” category, all 7 DAPs were up-regulated. However, the distribution of up-regulated and down-regulated DAPs within the “Plant hormone signal transduction” category was not specified (Figure 5B).

GO function analysis annotated 594 differential genes, out of which 62 were included in the top 30 entries, divided into three categories: molecular function, cell composition, and biological process. From a molecular function perspective, the largest proportion is associated with catalytic activity, specifically ”Binding” and ”Catalytic activity,” which contain 38 and 20 differential genes, respectively. The main functional locations of the differentially expressed genes are distributed in the organelle (“Organelle”) and the membrane system (“Membrane”). These differential genes are primarily involved in the biological processes of the ”Cellular process”, ”Metabolic process”, and ”Developmental process,” which include 43, 39, and 23 differential genes, respectively (Figure 6A). KEGG enrichment analysis revealed 51 differential genes among the top 20 enriched entries. Among them, particular attention was given to ”Plant hormone signal transduction” (CMO:04075), which contains 6 differential genes, and ”Cysteine methionine metabolism” (CMO:00270), which contains 2 differential genes. Based on the functional enrichment analysis of differential proteins and genes, it can be inferred that transcriptional regulation occurring in the nucleus plays a significant role in the female regulatory pathway of melon, with transcription factors containing zinc ions being relatively important. Additionally, plant hormone synthesis and signal transduction processes are closely related to female differentiation, particularly the significance of ethylene. Ubiquitination-mediated metabolism is also important (Figure 6B).

### 2.5. Differential Protein and Differential Gene Interaction Network Regulates Female Differentiation in Melon

Through functional enrichment analysis of 551 differentially abundant proteins and 594 differentially expressed genes, we have made preliminary inferences about the relevant pathways involved in the regulation of female development. The interactions between these differentially expressed proteins and genes within these pathways are believed to play a role in the regulation of female development. In this study, we performed an interaction network analysis of the differentially expressed proteins and genes potentially involved in sex regulation to investigate their interaction relationships.

In this study, 38 DAPs and 43 DEGs were screened to identify their potential involvement in the sex-determining pathway (Appendix A). After performing a functional enrichment analysis, it was found that these 81 DAPs/DEGs were primarily associated with ethylene biosynthesis, regulation of flower organ development, plant hormone signaling, and ubiquitinated protein metabolism. These four biological process modules formed a sex regulatory network (Figure 7A). Ethylene, which plays a crucial role in female differentiation, was found to be influenced by the up-regulation of eight DAPs and seven DEGs involved in the ethylene synthesis pathway (Figure 7B,C). Notably, SAH (MELO3C020432.2.1), CMT3 (MELO3C015649.2.1), and MET4 (MELO3C026448.2.1) were found to interact with multiple DAPs and DEGs. Additionally, two ACS genes (ACS7 and ACS11) and two ACO genes (ACO1 and ACO2) exhibited high connectivity in the network (Figure 7A). The interaction network revealed the significant roles of the ethylene receptor ETR (MELO3C003906.2.1) and MAPK protein kinase (MELO3C005705.2.1) in plant hormone signaling. These proteins were crucial intermediaries between ethylene synthesis and flower organ development (Figure 7A). They directly or indirectly promoted the expression of DL (MELO3C034075.2.1) and HEC3 (MELO3C008047.2.1), which are associated with carpel development. Notably, DL exhibited high expression levels among all the network DEGs. Moreover, ETR and MAPK indirectly inhibited the expression of another development-related gene, TGA9.1 (MELO3C021427.2.1) and TGA9.2 (MELO3C004728.2.1), through flower allotypes PMADS2 (MELO3C010515.2.1) and DEFA (MELO3C003778.2.1). Among all the network DEGs, TGA9.2 displayed lower expression levels (Figure 7B,C). Furthermore, it was observed that almost all proteins related to plant hormone signaling were directly or indirectly associated with the ubiquitinated protein degradation pathway. This finding suggests that the ubiquitinated protein degradation pathway plays a crucial role in hormone signaling and the melon sex-determining pathway.

### 2.6. Ethylene-Mediated Study of Female Differentiation Regulatory Pathways in Melon

According to omics analysis and previous research reports, this study proves that the ethylene-mediated sex determination pathway plays a key regulatory role in female development. A regulatory pathway model for melon female development is proposed based on functional enrichment analysis and interaction network analysis of differential proteins and differential genes.

As shown in Figure 8, ethylene is synthesized de novo with methionine (L-Methionine) as a substrate. S-adenosylmethionine synthionine synthase catalyzes the formation of S-adenosylmethionine (S-AdoMet). Part of S-AdoMet continues to enter the ethylene synthesis pathway, while another part enters the methionine cycle to continue synthesizing methionine. In the methionine cycle, two key enzymes catalyze the process: (cytosine-5)-methyltransferase (CMT; MELO3C026448.2.1, and MELO3C015649.2.1) and adenosine homocysteinase (AdoHS; MELO3C020432.2.1). Both enzymes have higher expression levels of protein (FvsM) and therefore contribute to more ethylene synthesis substrate (L-Methionine) in the female phenotype (F). S-adenosylmethionine is then catalyzed by ACS to form ACC (an ethylene precursor). There are four ACS-coding genes (MELO3C015444.2.1, MELO3C010779.2.1, MELO3C005597.2.1, and MELO3C006840.2.1) that are highly expressed (FvsM) to promote the production of more ethylene precursors. ACC is subsequently decomposed by ACO (ACC oxidase) to produce ethylene molecules. At this stage, three ACO-coding genes (MELO3C006437.2.1, MELO3C019735.2, and MELO3C014437.2) exhibit up-regulated expression (FvsM), catalyzing the production of more ethylene molecules. Ethylene acts as a small gas molecule for signaling by binding to ethylene receptors in the membrane system. The female phenotype has more ethylene receptors (MELO3C003906.2.1) capable of binding to C_2_H_4_ molecules (Figure 8). Ethylene signaling is mediated by MPK6, EIN2, and EIN3. The expression level of the MPK6 (MELO3C005705.2.1) protein is high in the female phenotype, and EIN3 is inhibited by EBF1/2 in this pathway. At this stage, the auxin signaling mechanism plays an important role. A series of auxin response proteins (ARPs) (AUX/IAA, GH3, and SAUR) can inhibit EBF1/2 [31]. In females, four auxin response protein-coding genes (MELO3C011268.2.1, MELO3C002378.2.1, MELO3C020765.2.1, and MELO3C013403.2) show high transcription levels, weakening the inhibitory effect of EBF1/2 on EIN3 in female manifestations. Auxin response factor (ARF) plays a crucial signaling role by producing ARPs that not only inhibit EBF1/2 but also directly up-regulate ACS expression [32]. In the female phenotype, two ARFs (MELO3C006181.2.1 and MELO3C023883.2.1) have extremely high protein abundance levels (Figure 8). Ethylene undergoes a signaling process that activates the expression of downstream ethylene response factor (ERF), which can promote the auxin signaling process. In addition, the ERF ultimately promotes melon female development through the regulation of downstream target genes. Six highly expressed ERFs (MELO3C015543.2.1, MELO3C026158.2.1, MELO3C005940.2.1, MELO3C006430.2.1, MELO3C006149.2.1, and MELO3C022358.2.1) may have important regulatory roles in this process.

Based on the above results, we finally proposed a pathway model for female regulation in melon. The model starts with the de novo synthesis of ethylene, which undergoes signal transduction and polyhormonal co-regulation of downstream target genes, ultimately leading to female sexual differentiation (Figure 8).

## 3. Discussion

### 3.1. Candidate Genes for Melon Female Development Revealed by Omics

Mining key genes involved in sex determination has always been a top priority in melon sexual differentiation research. For many years, researchers have been searching for sex-determining genes in melons using forward and reverse genetic approaches. While genetic analysis and genetic mapping have the advantages of high accuracy and strong relevance, they are often limited by the presence of mutants in the population. This is where the advantages of functional omics can be better utilized. Functional omics examines the differences between different sexes comprehensively and explores sex-determining genes and proteins through comparative studies involving sex-determining pathways. In this study, the differences between female and male phenotypes of melons were systematically analyzed at the multi-omics level, including the proteome and transcriptome. Through integrated comparative analysis, several important genes related to female development were discovered.

The study identified 551 DAPs and 594 DEGs involved in multiple pathways of melon growth and development. Based on functional annotation and enrichment analysis, four biological process modules closely related to female development were summarized: ethylene biosynthesis, flower organ development, plant hormone signaling, and ubiquitinated protein metabolism. These modules included 38 DAPs and 43 DEGs. Ethylene, a gaseous plant hormone that regulates sex differentiation in melon [3,33,34], plays a crucial role in female flower development. Therefore, two genes coding for cytosine-specific methyltransferase (CMT) (*MELO3C026448.2.1* and *MELO3C015649.2.1*), one gene coding for Adenosylhomocysteinase (AdoHS) (*MELO3C020432.2.1*), four genes coding for ACS (*MELO3C015444.2.1*, *MELO3C010779.2.1*, *MELO3C005597.2.1*, and *MELO3C006840.2.1*), and three genes coding for 1-aminocyclopropane-1-carboxylate oxidase homolog 1-like (ACO) (*MELO3C006437.2.1, MELO3C019735.2*, and *MELO3C014437.2*) are considered important candidate genes. Among these, two ACS genes (*MELO3C015444.2.1* and *MELO3C010779.2.1*) have been identified as melon sex-determining genes [21,22]. Additionally, an ACO-coding gene (*MELO3C007425.2.1*) has been reported to be associated with melon female development. Hence, the importance of the two ACS genes and three ACO-coding genes in this study is further highlighted.

Ethylene promotes female development through hormone signaling pathways, making the high-expression protein MPK6 coding gene (*MELO3C005705.2.1*) a significant candidate for female regulation in this process. Interactive protein network analysis has shown crosstalk clues between different phytohormones (Figure 7). Among these, there is a high degree of connection between ethylene and auxin, gibberellin (GA), methyl jasmonate (MeJA), and other signaling pathways. Many studies have indicated a mutual promotional relationship between ethylene and auxin, particularly during signal conduction. ARF can directly and positively regulate the expression of ACS, while ARP can indirectly inhibit EBF1/2, thereby promoting ethylene signaling [11,35,36]. Therefore, in this process, two ARF-coding genes (*MELO3C006181.2.1* and *MELO3C023883.2.1*) and four ARP-coding genes (*MELO3C011268.2.1, MELO3C002378.2.1, MELO3C020765.2.1,* and *MELO3C013403.2.1*) are also important candidates. Among them, the positive regulation of the two ARFs aligns more closely with the mechanism of ‘Y101′ dominant strong female regulation. As a gaseous phytohormone, ethylene plays a crucial role in regulating the development of flower organs and acts as a significant mediator in the ethylene response factor (ERF) regulation mechanism. Previous studies have demonstrated that *CsERF110* in cucumbers can directly bind to *CsACS11*, activating its expression to promote female development. This finding has also been confirmed in melons [34]. Additionally, *CsERF31* has been shown to enhance cucumber female differentiation [37]. Based on these findings, we hypothesize that the six highly expressed ERFs identified in this study, namely *MELO3C015543.2.1*, *MELO3C026158.2.1*, *MELO3C005940.2.1*, *MELO3C006430.2.1*, *MELO3C006149.2.1*, and *MELO3C022358.2.1*, may have a regulatory effect on melon female differentiation.

### 3.2. A Regulatory Network Model for the Female Development of Melon

The hermaphroditic flower type material ‘Y101’ is a typical female phenotype and a wild-type thin-skinned melon. We conducted comparative omics analysis of the female pool (F) and male pool (M) constructed by the melon material F_2_ population by ‘Y101’ and the male phenotype ‘0426’. This approach helped us eliminate background errors between species.

A series of candidate genes that may have a role in female flower development were revealed by functional enrichment analysis and network interaction analysis. Results indicated that ethylene pathway genes play an essential role in determining female flower development. Through omics analysis, the study speculates that the enhanced expression of *CmACS7* and *CmACS11* may also be regulated by ARF. Additionally, the study shows that ‘Y101’ exhibits a high expression of (cytosine-5)-methyltransferase (CMT) and adenosine homocysteine (SAH), leading to an abundance of ethylene synthesis substrates. With the high expression of the ethylene precursor ACC oxidase (ACO), the flower buds of ‘Y101’ contain more ethylene during development. Sufficient ethylene activates a series of ERF expressions through the signaling process, ultimately promoting female development by regulating downstream target genes. Network interaction analysis suggests that these downstream target genes of ERF may be related to the development of female organ-related genes, including *CRABS CLAW* (MELO3C034075.2.1), *HECATE3* (*MELO3C008047.2.1*), and *MADS13* (*MELO3C033521.2.1*). An interesting fact identified in previous studies is that the majority of the genes associated with sex differentiation are ethylene synthase genes [3]. Our study proposes a putative model (see Figure 8) that further supports the hypothesis that the ethylene synthase network plays a crucial role in melon, regulating sex differentiation. A comparison of female (F) and male (M) pools in our study has uncovered several candidate genes within the ethylene signaling pathway.

As illustrated in Figure 8, four catalytic enzymes, including ACO, ACS, CMT, and SAH-related genes and proteins, exhibited a multifold increase. Additionally, proteins encoding AUX/IAA were up-regulated, suggesting the involvement of more genes in female enhancement. It is noteworthy that only selective genes were studied in previous experiments, and Li et al. (2019) recommended, in their review on gene interactions regulating sex determination, conducting expression regulation studies for each gene to obtain clearer evidence of sex differentiation in cucurbits [3]. Interestingly, an integrated network analysis of key DAPs and DESs from our comparative groups highlighted four interaction networks, including those involving plant hormones, transduction, and ethylene biosynthesis genes. Our proposed model indicates that auxin-related genes, such as ARPs and ARFs, also exhibited higher expression, suggesting their potential role in sex differentiation.

It is well established in previous literature that auxins promote ethylene biosynthesis by up-regulating ACS, the first regulatory step in ethylene biosynthesis, and our findings support these hypotheses. Similarly, several auxin-responsive factors were differentially expressed in our study. The proteome analysis also revealed that two key ARFs (ME-LO3C023883.2.1 and MELO3C006181.2.1) showed a two-fold increase in the female dominant melon. The female regulatory candidate genes and regulatory networks identified through omics analysis require further verification. Nonetheless, the current research results hold important reference value and have significant implications for in-depth research. It is believed that with the improvement of the melon transformation system and further research, the regulatory mechanism of sex determination in melons can be better understood in the future. The current study encompasses various aspects, including transcriptomics, proteomics, and the generation of sex type-specific pools for research. A detailed explanation and analysis of all crucial pathways in terms of female development provide essential information for researchers. However, the omics data presented in the study are complex and high-dimensional, posing challenges for analysis and interpretation. Furthermore, the study unveiled new candidates that require further investigation to validate their true function in melon female development. The candidate genes and proteins identified in the current study can be utilized in other cucurbits as a hypothesis to identify the sex differentiation mechanism.

## 4. Materials and Methods

### 4.1. Plant Material and Flower Bud Collection

The plant monoecious line ‘0426’ (ssp. *melo*) and hermaphrodite line ‘Y101’ (ssp. *agrestis*) were grown in the cucurbit research and experimental station, College of Horticulture, Northwest A & F University, Yangling, Shaanxi, China [26]. Plants of ‘0426’ were used as the female parent and ‘Y101’ as the male parent to obtain the F_1_ (0426 × Y101) genetic population. F_2_ plants were obtained by selfing F_1_ plants in the greenhouse. The isolated population of F_2_ (0426 × Y101) contained four sex types of individuals: gynoecy, hermaphrodite, monoecy, and andromonoecy [26]. The gynoecy and hermaphrodite plants were considered female sexual types and kept under the F-pool [38,39]. The flower buds collected from monoecy and andromonoecy plants are considered male-dominant plants and grouped under the M-pool [38,39]. At the 15-node plant growth stage, when sex type judgment is reliable, 50 gynoecious plants and 50 hermaphrodite plants were selected from the isolated group, and the apex tissue was collected to construct the F (female) pool (Figure 9). Similarly, 50 each of monoecious plants and andromonoecious plants were selected, and the apex tissue was sampled to construct the M (male) pool. Samples were quickly placed in liquid nitrogen after collection, and whole proteins and RNA were extracted for proteome and transcriptome sequencing.

### 4.2. TMT Labeling Quantitative Proteome Sequencing and Analysis

The samples were collected and then mixed in a single pool before being sent to Shanghai Luming Biotechnology Co., Ltd. (Shanghai, China) for protein detection and comparative analysis of the F and M-pool samples [40,41]. The main experimental process involved extracting the total protein from the F pool and the M pool, which was a mixture of collected apex tissue. A small amount of the extracted total protein was taken for polyacrylamide gel electrophoresis (SDS-PAGE) to detect protein quality and integrity. Once the protein bands were successfully detected on the gel, trypsin was used to enzymatically hydrolyze the total protein, breaking it down into peptides. The peptides were then labeled using TMT reagents. TMT (6 standard) labeled peptides were separated using reversed-phase chromatography on an Agilent 1100 HPLC liquid chromatograph (column: Agilent Zorbax Extend—C18 narrow bore column, 2.1 × 150 mm, 5 μm). Finally, LC-MS analysis was performed on the separated samples to obtain the raw data. The data were compared with the melon reference proteome data (http://cucurbitgenomics.org/v2/ftp/genome/melon/DHL92/v4.0/, accessed on 16 September 2023) for qualitative and quantitative analysis. Quality control was conducted on the searched data, followed by expression level analysis and functional analysis. The false discovery rate (FDR) was adjusted to <1%, and the minimum score for peptides was set at >40. Differentially abundant proteins (DAPs) were identified by performing a Student’s *t*-test with a fold change >1.5 (or <0.67) and a *p*-value <0.05, comparing them to the healthy control [42,43]. Additionally, Fisher’s exact test was used to test DAPs against the background of the identified proteins.

### 4.3. F/M Mixed Pool RNA-Seq and Parametric Transcriptome Analysis

The bud samples were collected and sent to Shanghai Luming Biotechnology Co., Ltd. for RNA-seq and comparative analysis of the F pool and M pool samples. The main steps followed for comparative transcriptome analysis included extracting total RNA from the apex tissue mixed in the F pool and M pool and constructing the cDNA library after assessing the purity, integrity, and concentration of the RNA sample. In this study, total RNA extraction was carried out utilizing the TIANGEN RNAprep Pure kit (DP432) from Tiangen Biotech Co., Ltd. (Beijing, China). Following the manufacturer’s instructions, apex tissue of melon varieties was collected, and any specified pre-processing steps, such as homogenization or cell lysis, were performed accordingly. The RNA extraction protocol provided by the kit was rigorously followed, incorporating specific incubation times and temperatures. Purification was executed according to the provided guidelines, utilizing recommended columns or membranes. The elution step involved the use of RNase-free water or elution buffer, following the manufacturer’s instructions for optimal recovery. To assess the quality of the extracted RNA, measurements of purity and concentration were performed using NanoDrop (Agilent Technologies, Santa Clara, USA), providing key parameters such as the A260/A280 ratio. Furthermore, the integrity of the RNA was evaluated using agarose gel electrophoresis, ensuring the suitability of the extracted RNA for downstream applications. The Illumina HiSeq platform was used to bilaterally sequence the library and obtain the raw data. After performing quality control on the original sequencing data (reads), the final clean reads were summarized. The obtained sequence data were then compared with the melon reference genome (DHL92, V3.6.1, http://cucurbitgenomics.org/v2/organism/23, accessed on 16 September 2023) to obtain information about the reads’ alignment to the reference genome. Reads with fewer than 10 comparisons, reads that aligned to multiple regions, and reads without alignment to the reference were subsequently removed. Finally, the quantitative expression level of each gene was obtained. This was followed by an analysis of the expression levels and functional annotations.

### 4.4. Functional Enrichment of DEGs and DAPs

The GO enrichment analysis of the differentially expressed genes (DEGs) and differentially abundant proteins (DAPs) was performed using the GOseq R package, which is based on the Wallenius non-central hypergeometric distribution [44]. This approach allows for the adjustment of gene length bias in DEGs and DAPs. The GO annotation proteome used in the analysis was obtained from the UniProt-GOA database (http://www.ebi.ac.uk/GOA/, accessed on 16 September 2023). To test the statistical enrichment of DEGs and DAPs in KEGG pathways, we utilized the KOBAS V3.0 software (http://bioinfo.org/kobas/download/, accessed on 16 September 2023) [45].

### 4.5. Protein Interaction Network Analysis

According to the analysis results of the proteome and transcriptome, relevant proteins and genes that may be involved in the sex-determining pathway were screened from the KEGG-enriched DAPs and DEGs. To analyze the interactions among these proteins, an online protein interaction prediction and analysis software, STRING (https://cn.string-db.org/, accessed on 16 September 2023), was employed. Furthermore, Cytoscape V3.7.2 software was utilized to map the protein–protein interaction (PPI) networks.

### 4.6. qRT-PCR Analysis of Omics Data and Statistical Analysis

To estimate the transcript levels of selected differentially enriched proteins and genes, qRT-PCR was conducted using terminal bud tissue samples that were previously used for proteomics and RNA-seq. Twenty genes, both up-regulated and down-regulated, were randomly chosen from the proteome and transcriptome results to validate the reliability of the sequencing data. The cDNA synthesis was carried out using two micrograms of total RNA with SuperScript IV reverse transcriptase (Invitrogen) and an oligo(dT) primer. The qRT-PCR reaction (25 μL) included 1 μL of the cDNA template, 0.5 μL of each primer, 12.5 μL of 2 × SYBR qPCR Mix, and 11 μL ddH_2_O to a final volume of 25 μL. Three technical replicates per sample set were performed. qRT-PCR amplification conditions were set to 94 °C for 3 min, followed by 40 cycles of 94 °C for 10 s, and 60 °C for 30 s. The qRT-PCR was performed on a BIO-RAD CFX96 machine using gene-specific forward and reverse primers (Appendix A). We also examined the transcript levels of selected DEGs and DAPs in the female type parent (Y101) and male type parent (0426) to additionally analyze and confirm the dependability of the sequencing data and the potential functionalities in sex regulation.

The data were analyzed using the 2^−ΔΔCT^ method, and *CmActin* was selected as the reference gene for normalization. Significance analysis was performed using IBM SPSS 19.0 software for ANOVA, and the Student’s *t*-test was employed for further analysis.

## 5. Conclusions

In the present study, we performed transcriptome analysis followed by proteome analysis of two different sex type pools, including flower buds of male and female characteristics. A high number of DEGs and DAPs were identified by proteome and transcriptome analysis of the female pool (F) and male pool (M) identified by the segregated population in the F_2_ generation. Through functional enrichment analysis, 38 DAPs and 43 DEGs were identified, which were involved in the regulatory network of female flower development. Through systematic analysis, the female developmental regulatory pathway model of ethylene biosynthesis, signal transduction, and target gene regulation was proposed. Two CMTs (*MELO3C026448.2.1* and *MELO3C015649.2.1*), one AdoHS, four ACSs (*MELO3C015444.2.1*, *MELO3C010779.2.1*, *MELO3C005597.2.1*, and *MELO3C006840.2.1*), three ACOs (*MELO3C006437.2.1*, *MELO3C019735.2*, and *MELO3C014437.2*), two ARFs (*MELO3C006181.2.1* and *MELO3C023883.2.1*), four ARPs (*MELO3C011268.2.1*, *MELO3C002378.2.1*, *MELO3C020765.2.1*, and *MELO3C013403.2.1*), and six ERFs (*MELO3C015543.2.1*, *MELO3C026158.2.1*, *MELO3C005940.2.1*, *MELO3C006430.2.1*, *MELO3C006149.2.1*, and *MELO3C022358.2.1*) were found to have an important role in pathways associated with female flower development. Future investigations into the regulatory mechanisms of some newly identified candidate genes and proteins may enable researchers to manipulate specific sexual expressions in melon. The functional annotation of high-throughput transcriptomic and proteomics data generated in this study may serve as genetic resources for the development of crop improvement strategies for melon and various other cucurbits.

## Figures and Tables

**Figure 1 ijms-24-16905-f001:**
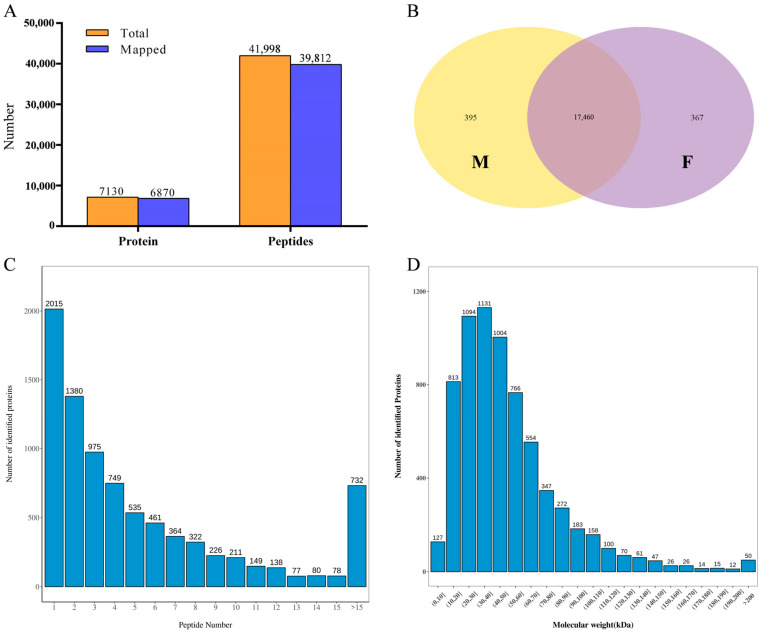
Overview of the multi-omics data. (**A**) Protein and peptide identification information; (**B**) Venn diagram of transcriptome gene identification; (**C**) Statistics of protein peptides; (**D**) Protein molecular mass distribution map (kDa).

**Figure 2 ijms-24-16905-f002:**
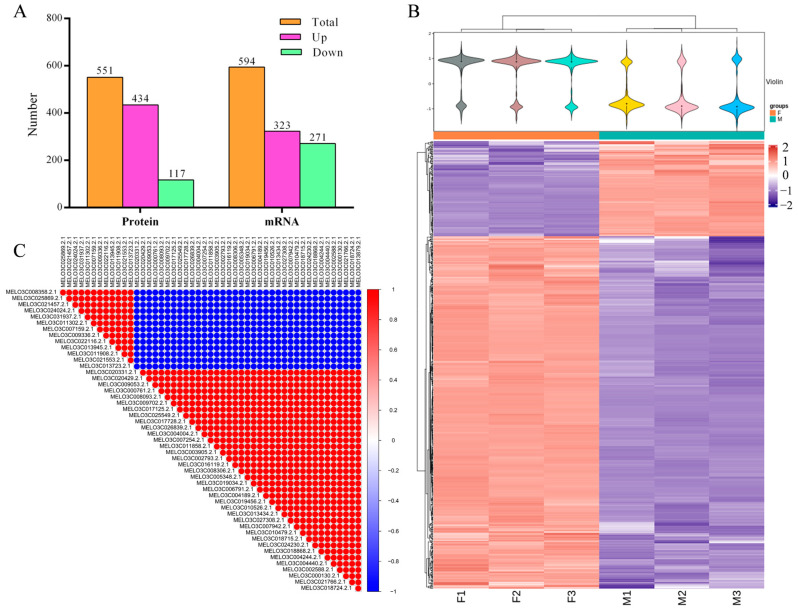
Differential protein/gene identification and analysis. (**A**) Differential protein and differential gene statistics; (**B**) Heat map of differential protein clustering of female (F1, F2, and F3) and male (M1, M2, and M3) pools; (**C**) Correlation analysis of the expression of the top 50 DAPs, with the color bar representing the correlation coefficient.

**Figure 3 ijms-24-16905-f003:**
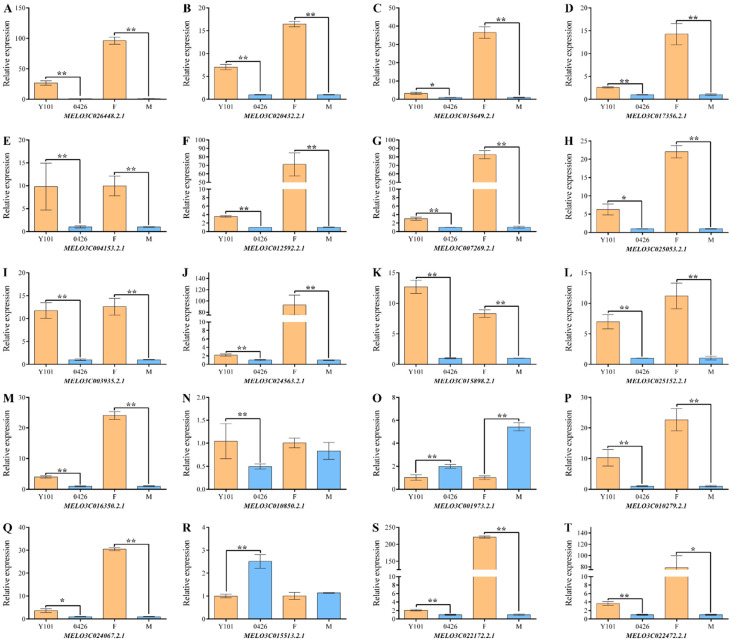
Identification of transcription levels of differential proteins. The DAPs that are down-regulated include (**N**,**O**,**R**), whereas those that are up-regulated include the remaining set. (**A**–**E**) ethylene synthesis-related protein; (**F**) gibberellin-associated protein; (**G**) auxin-associated protein; (**H**,**I**) proteins associated with flower organ development; (**J**–**O**) phytohormone signaling and transduction; (**P**) sexual cycle protein; (**Q**,**R**) transcription factors that regulate growth and development; (**S**,**T**) membrane proteins. The relative expression (*Y*-axis) was calculated using the 2^ΔΔCt^ method, and melon varieties (‘Y101’ and ‘0426’) with F and M pools are shown on the *X*-axis. Bar graphs are plotted with the mean ± SD of three replications. * and ** represent significant differences in expression levels at *p* < 0.05 and *p* < 0.01, respectively (Student’s *t*-test).

**Figure 4 ijms-24-16905-f004:**
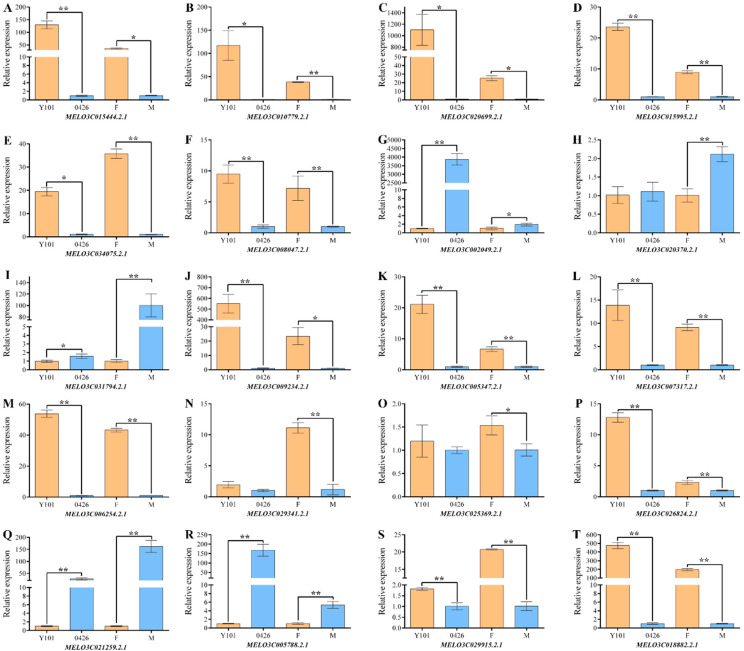
qRT-PCR verification of differential genes. The DEGs that are down-regulated include (**G**–**I**,**Q**,**R**), whereas those that are up-regulated include the remaining set. (**A**,**B**) ethylene synthesis-related proteins; (**C**) gibberellin-associated protein; (**D**) auxin-associated protein; (**E**–**I**) proteins associated with flower organ development; (**J**,**K**) zinc finger proteins; (**L**) transmembrane protein; (**M**) membrane proteins; (**N**–**R**) enzymes; (**S**,**T**) unknown proteins. The relative expression (*Y*-axis) was calculated using the 2^ΔΔCt^ method, and melon varieties (‘Y101’ and ‘0426’) with F and M pools are shown on the *X*-axis. Bar graphs are plotted with the mean ± SD of three replications. * and ** represent significant differences in expression levels at *p* < 0.05 and *p* < 0.01, respectively (Student’s *t*-test).

**Figure 5 ijms-24-16905-f005:**
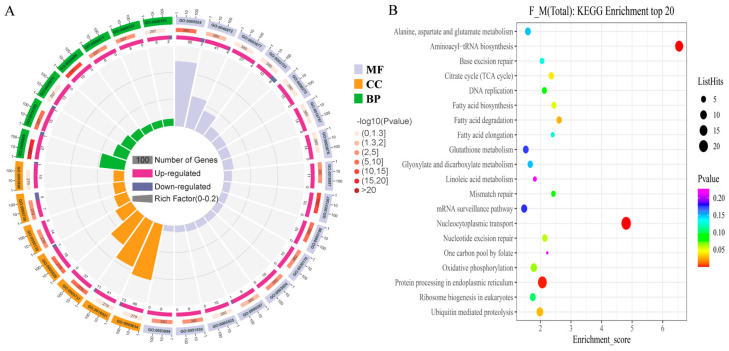
GO and KEGG enrichment analyses of differential proteins. (**A**) TOP30 information under GO item (GO entries with the number of corresponding differential proteins greater than 1 in the three categories were screened, and 10 entries were sorted from largest to smallest according to the −log10pvalue corresponding to each entry). MF: molecular function; CC: component; BP: biological process. (**B**) TOP20 KEEG enrichment information (each entry is sorted from largest to smallest by its −log10pvalue). The dot size represents the number of DAPs under this enriched item.

**Figure 6 ijms-24-16905-f006:**
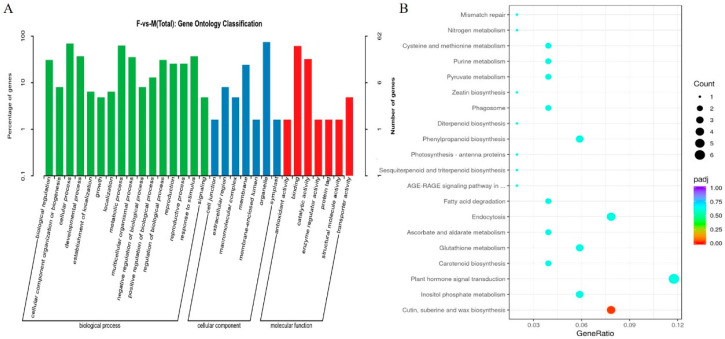
GO and KEGG enrichment analyses of differential genes. (**A**) TOP30 information under the GO item (GO entries with the number of corresponding differential proteins greater than 1 in the three categories were screened, and 10 entries were sorted from largest to smallest according to the −log10pvalue corresponding to each entry). (**B**) TOP20 KEEG enrichment information (each entry is sorted from largest to smallest by its −log10pvalue). The dot size represents the number of DEGs under the enriched item.

**Figure 7 ijms-24-16905-f007:**
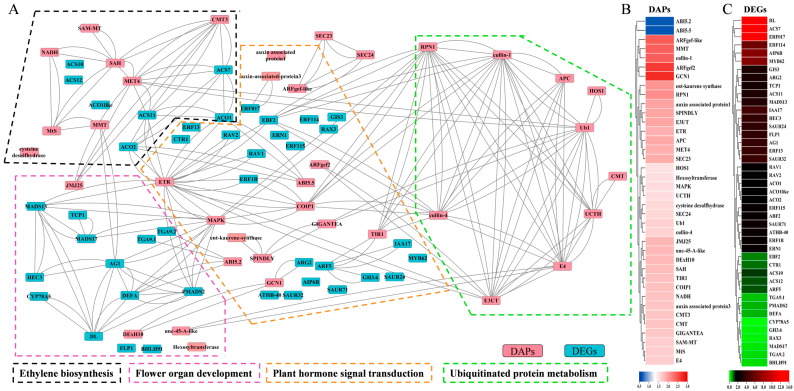
Integrated network analysis of key DAPs and DEGs. (**A**) Interaction network of key DAPs and DEGs involved in ethylene synthesis, floral organ development, plant hormone signaling, and ubiquitinated protein metabolism. The nodes represent DAPs or DEGs, the solid lines represent interaction relationships, and the dashed lines in different colors represent biological process modules. (**B**,**C**) Heat maps of DAP and DEG expression in the network. The color bars are fold-change values.

**Figure 8 ijms-24-16905-f008:**
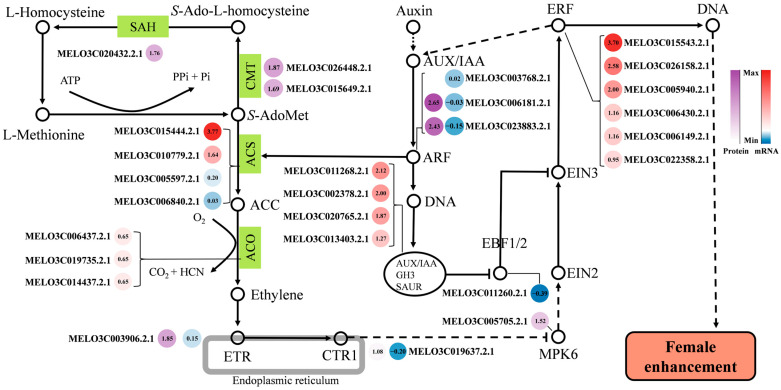
Ethylene-mediated regulation of female development in melon. Hollow dots represent molecules or products, green squares represent catalytic enzymes at that stage, sharp lines represent synthesis or promotion, T-shaped lines represent inhibition, dashed lines represent the presence of other processes, different colored origin points represent DAPs or DEGs, and protein color bars represent fold-change values of DAPs. mRNA color bars represent Log_2_ fold-change values of DEGs.

**Figure 9 ijms-24-16905-f009:**
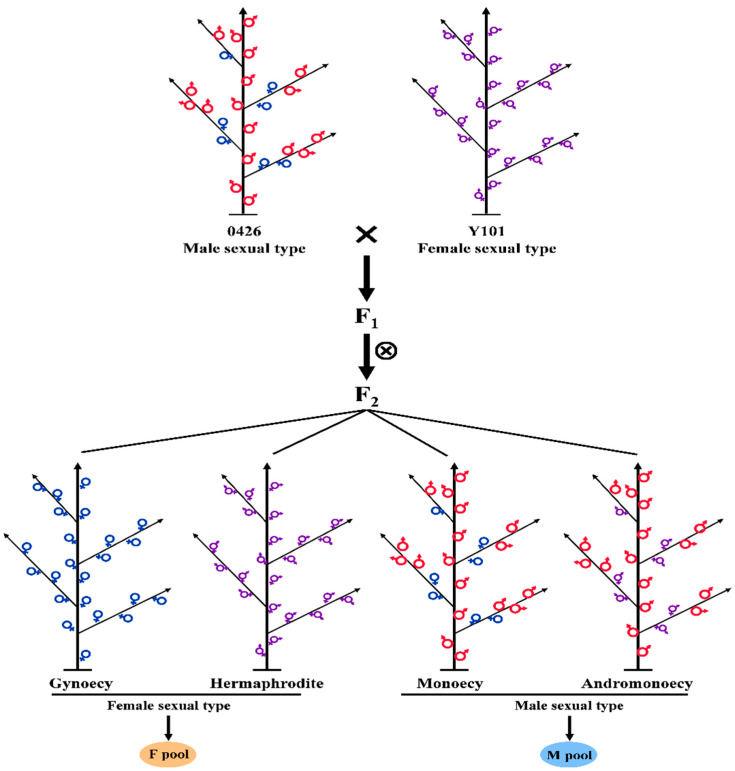
Representation of sexual phenotypes and female (F) and male (M) pools of plant materials used in the current study. Monoecious line ‘0426’ (ssp. *melo*) and hermaphrodite line ‘Y101’ (ssp. *agrestis*) were used as parents, and the F/M pool was constructed using the F_2_ population.

**Table 1 ijms-24-16905-t001:** Overview of the transcriptome data.

Pool	Total	Total Map	Unique Map	Multimap	Positive	Negative
F	45,179,104	43,344,617 (95.94%)	42,839,380 (94.82%)	505,237 (1.12%)	21,413,961 (47.4%)	21,425,419 (47.42%)
M	44,568,262	42,604,523 (95.59%)	42,122,518 (94.51%)	482,005 (1.08%)	2,1056,584 (47.25%)	2,1065,934 (47.27%)

## Data Availability

Data is contained within the article and Appendix A.

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
