# Peer review of "Multi-Omics Analysis Reveals Intricate Gene Networks Involved in Female Development in Melon"

_ijms, 2023, doi:10.3390/ijms242316905_

Round 1
Reviewer 1 Report
Comments and Suggestions for Authors
In this study, the authors identified intricate gene networks involved in female development in melon using multi-omics analysis. The authors discovered that four biological process modules, including ethylene biosynthesis, flower organ development, plant hormone signaling, and ubiquitinated protein metabolism, are related to female development in melon. The manuscript is well-written and presented. However, before accepting this paper, the following minor issues should be resolved:
Line 18: Please use the full name of DAPs (Differentially Abundant Proteins) and DEGs (Differentially Expressed Genes) before using the abbreviation.
Lines 500-501: Please provide information on the environmental conditions in which the melon plants were grown.
Discussion: The authors should discuss the limitations of the current study.
Author Response
Dear reviewer,
Thank you for reviewing our manuscript titled " Multi-omics analysis reveals intricate gene networks involved in female development in melon " submitted to IJMS. Your comments have been invaluable in improving the quality and rigor of our work. Based on your suggestions, we have made changes to the manuscript. We look forward to your evaluation of the updated version. We have carefully considered each of your suggestions and concerns, and we provide our responses below(PDF:Response to Reviewer1):
Sincerely,

Reviewer 2 Report
Comments and Suggestions for Authors
Kindly find the comments at the attachment.

Comments on the Quality of English Language
Minor editing of English language required
Author Response
Dear Reviewer,
We would like to express our gratitude for your thoughtful and constructive feedback on our manuscript titled " multi-omics analysis reveals intricate gene networks involved in female development in
melon " submitted to IJMS. Your comments have been invaluable in improving the quality and rigor of our work. We hope that these revisions address your concerns effectively. If there are any remaining issues or if you require further clarification on specific points, please do not hesitate to let us know. We would like to take this opportunity to acknowledge your expertise and guidance. Your feedback has undoubtedly improved the overall quality of our research findings. We look forward to your evaluation of the updated version. We have carefully considered each of your suggestions and concerns, and we provide our responses below (PDF: Response to Reviewer2):
Sincerely,

Round 2
Reviewer 2 Report
Comments and Suggestions for Authors
Kindly find the comments in the attachment.

Comments on the Quality of English Language
Minor editing of English language required
Author Response
Dear Reviewer,
Thank you again for reviewing our revised manuscript (ijms-2684353-review, " multi-omics analysis reveals intricate gene networks involved in female development in melon "). Your suggestions for improving the quality of our manuscripts are extremely important. We have carefully considered each of your suggestions and concerns, and we provide our responses below (PDF file):
Sincerely,
